# Comparative Pathogenomic Analysis of Two Banana Pathogenic *Dickeya* Strains Isolated from China and the Philippines

**DOI:** 10.3390/ijms232112758

**Published:** 2022-10-22

**Authors:** Chenxing Tan, Chuhao Li, Ming Hu, Anqun Hu, Yang Xue, Xiaofan Zhou, Jianuan Zhou

**Affiliations:** Guangdong Province Key Laboratory of Microbial Signals and Disease Control, Integrative Microbiology Research Centre, South China Agricultural University, Guangzhou 510642, China

**Keywords:** banana soft rot, *Dickeya*, virulence factors, genomic comparison

## Abstract

*Dickeya* is a major and typical member of soft rot Pectobacteriaceae (SRP) with a wide range of plant hosts worldwide. Previous studies have identified *D. zeae* as the causal agent of banana soft rot disease in China. In 2017, we obtained banana soft rot pathogen strain FZ06 from the Philippines. Genome sequencing and analysis indicated that FZ06 can be classified as *D. dadantii* and represents a novel subspecies of *D. dadantii*, which we propose to name as subsp. *paradisiaca*. Compared with Chinese banana soft rot pathogenic strain *D. zeae* MS2, strain FZ06 has a similar host range but different virulence; FZ06 is significantly less virulent to banana and potato but more virulent to Chinese cabbage and onion. Characterization of virulence factors revealed obviously less production of pectate lyases (Pels), polygalacturonases (Pehs), proteases (Prts), and extrapolysaccharides (EPSs), as well as lower swimming and swarming motility and biofilm formation in strain FZ06. Genomic comparison of the two strains revealed five extra gene clusters in FZ06, including one Stt-type T2SS, three T4SSs, and one T4P. Expression of cell wall degrading enzyme (CWDE)-encoding genes is significantly lower in FZ06 than in MS2.

## 1. Introduction

Banana (*Musa* spp.) is native to tropical Asia and is an important cash crop. China has a long history of banana cultivation and ranks second in global banana production, with a planting area exceeding 400 thousand hectares [1]. Bacterial soft rot caused by *Dickeya* has threatened banana yield worldwide since the 1980s [2,3,4,5]. The genus now includes 12 species, including *D. chrysanthemi*, *D. dianthicola*, *D. dadantii*, *D. zeae* [6], *D. solani* [7], *D. aquatic**a* [8], *D. fangzhongdai* [9], *D. undicola* [10], *D. lacustris* [11], *D. poaceaephila* [12], *D. oryzae* [13], and *D. parazeae* [14]. In addition, *D. dadantii* includes a subspecies, *D. dadantii* subsp. *dieffenbachiae* [15]. From the perspective of disease severity, *D. dadantii*, *D. solani*, *D. zeae*, and *D. fangzhongdai* cause considerable agricultural economic losses, among which only *D. dadantii* and *D. zeae* have been reported to naturally infect banana.

*D. dadantii* causes soft rot in a variety of crops and ornamentals, including potato, sweet potato, banana, maize, pineapple, and *Dianthus* spp. In 2006, bacterial stem and root rot of sweet potato caused by *D. dadantii* occurred in several provinces in China, becoming a quarantine plant disease that struck a devastating blow to the sweet potato industry in China [16]. Banana sheath rot caused by *D. dadantii* is a new banana disease that causing increasing damage since 2011 in Guangdong, Guangxi, Yunnan, Fujian, and Hainan provinces, with incidence rates of up to 70%–80% in some field plots [17]. *D. dadantii* 3937 isolated from *Streptocarpus ionanthus* is one of the most thoroughly studied members of *Dickeya* [18,19,20,21], for which many virulence factors have been characterized, such as plant-cell-wall-degrading enzymes (PCWDEs), extrapolysaccharides (EPS), siderophores, pigment indigoidine, type III secretion system (T3SS), flagellar and chemotaxis genes, and quorum-sensing (QS) systems [22,23,24,25,26].

*D. zeae* shows considerable differences in its physiological and biochemical traits from other species in *Dickeya* and has an extended host range [27]. It can naturally infect monocotyledons, such as maize, rice, banana, pineapple, *Brachiaria*, hyacinth, Clivia, *Canna indica*, and taro [5,21,28,29,30], as well as dicotyledons, such as potato, tobacco, and *Chrysanthemum* [6,31,32,33]. In recent years, *D. zeae* has been prevalent in China, causing serious soft rot on banana [4]. From 2010 to 2012, the average incidence increased from 20% to 70%, with up to 90% incidence in some plantations [34]. Studies on the virulence of *D. zeae* revealed similar patterns of pathogenic mechanism and repertoire of virulence factors in *D. dadantii* 3937, manifested as the AHL QS system and the cyclic-di-GMP second messenger regulating cell motility and biofilm formation [35,36,37,38], the VFM QS system modulating the production of PCWDEs [38,39], and the SlyA and Fis regulators regulating the virulence-associated phenotypes [40,41]. Characterization of *D. zeae* banana strain MS2 revealed an additional virulence factor that is a novel antibiotic-like toxin(s) [5,36] that differs from the polyketide polyamine antibiotic zeamines [42]. The structure of the toxin substance and the molecular mechanism of the interaction process with the host are not yet known.

In August 2018, we obtained pathogenic strain FZ06 isolated from diseased banana samples from the Philippines. In this study, we performed whole-genome sequencing and phylogenetic analysis of FZ06 and found that it can be classified as *D. dadantii* and represents a novel subspecies, which we propose to name as *D. dadantii* subsp. *paradisiaca*. In order to explore whether there is any pathogenic difference between the *Dickeya* strains isolated from Philippine and Chinese bananas, we examined phenotypic characteristics of the two strains (MS2 and FZ06), such as pathogenicity, virulence factor production, bacterial motility, biofilm formation, etc. Genomic comparison was also performed to investigate the pathogenic difference between the two strains at the genetic level.

## 2. Results

### 2.1. FZ06 and MS2 Have a Similar Host Range but Different Virulence on Various Hosts

Because FZ06 and MS2 were isolated from soft rot bananas in different locations, we compared their virulence-associated features at both phenotypic and genetic levels. First, we individually examined their growth dynamic processes in LB medium and found that they both reached their maximum bacterial density at 12~16 h, with comparable growth rates (Appendix A). To determine the host range and virulence of strain FZ06, we performed inoculation on 13 reported hosts of *D. zeae* MS2 [5]. The results showed that strain MS2 and strain FZ06 infected all the tested plants and that they have a similar host range but showed a divergence in terms of virulence. FZ06 was less virulent than MS2 on *Cucumis sativus*, *Solanum lycopersicum*, and *Musa* plants but more virulent on *Benincasa hispida*, *Brassica rapa* subsp. *pekinensis*, and *Allium cepa* (Appendix A, Table 1). The two strains had comparable virulence on *Raphanus sativus*, *Daucus carota*, *Lycopersicon esculentum*, *Solanum melongena*, *Zingiber officinale*, and *Colocasia esculenta* (Appendix A, Table 1).

### 2.2. FZ06 Exhibited a General Reduction in Various Virulence Traits

We measured multiple virulence factors produced by strains FZ06 and MS2 to determine differences in virulence between the strains. The results showed that the production of extracellular enzymes, including pectate lyases (Pel), polygalacturonases (Peh), and proteases (Prt), was lower in FZ06 than in MS2 (Figure 1). In addition, strain FZ06 did not produce detectable phytotoxin antagonistic to *E. coli* DH5α, whereas strain MS2 did (Figure 1). EPS is another virulence determinant in *Dickeya* pathogens. Similarly, compared with MS2, FZ06 produced significantly less EPS (Figure 2A). The formation of biofilms in FZ06 was also reduced compared with strain MS2 (Figure 2B). Motility of *Dickeya* pathogens usually includes swimming and swarming mediated by flagella, as well as twitching mediated by pili. Measurement of these phenotypes indicated that strain FZ06 is less motile than strain MS2, especially in swimming and swarming ability (Figure 2C).

### 2.3. FZ06 Was Identified as D. dadantii subsp. paradisiaca Based on Genomic Analysis

We sought to elucidate the molecular basis underlying the different pathogenicity-associated phenotypes between FZ06 and MS2. First, we sequenced the whole genome of FZ06. High-throughput genome sequencing of FZ06 generated 2.02 Gb (148,621 reads) of nanopore long-read sequencing data and 1.03 Gb (3.43 M pairs of PE150 reads) of BGISEQ short-read sequencing data. A circular 5,208,659 bp chromosome was obtained based on de novo assembly of nanopore long reads and subsequent polishing using BGISEQ short reads (Figure 3). The GC content of the FZ06 genome is 56.42%, which is slightly higher than that of the MS2 genome (53.45%) (Table 2). The genome contains 4396 protein-coding, 75 tRNA, 9 ncRNA, and 22 rRNA genes (Table 2).

Analysis of 16S rRNA sequences from the whole genome indicated that FZ06 was attributed to the genus *Dickeya*. To further determine the taxonomic status of strain FZ06, we performed ANI analysis on FZ06 and 167 *Dickeya* strains with genome sequence available in the NCBI RefSeq database. The ANI values between FZ06 and all the *Dickeya* genomes range from 81.88% to 98.50% (Appendix A). Notably, FZ06 shares more than 96% ANI values with all the 16 *D. dadantii* strains (Appendix A), which is higher than the species threshold of 95% [43]. Thus, FZ06 is classified as species *D. dadantii*.

To further investigate the relationships between FZ06 and other representative *Dickeya* strains, we then performed a phylogenetic analysis based on the 1264 single-copy orthologous genes shared by FZ06, all 16 *D. dadantii* strains, and type strains of 11 other *Dickeya* species. The relatedness analysis showed that FZ06 and the 16 *D. dadantii* strains form a monophyletic clade separate from other *Dickeya* species. Within the *D. dadantii* clade, FZ06 was most closely related to A622-S1-A17 and S3-1, which, together, were sister to *D. dadantii* subsp. *dieffenbachiae* NCPPB 2976 (Figure 4). Furthermore, FZ06 shares dDDH values above 79%, a commonly accepted boundary for subspecies, with A622-S1-A17 (86.5%) and S3-1 (85.2%) but not NCPPB 2976 (73.1%) (Appendix A). Therefore, our results indicate that strains FZ06, A622-S1-A17, and S3-1 represent a new subspecies of *D. dadantii*, which we propose to name as *D. dadantii* subsp. *paradisiaca*, with FZ06 as the type strain. 

### 2.4. Genome Comparison of Strains FZ06 and MS2

To elucidate the molecular basis underlying the different pathogenicity-associated phenotypes between FZ06 and MS2, we performed an in-depth genome comparison. Orthologous gene clusters were first constructed; then, FZ06 unique genes were obtained, annotated, and classified (Appendix A). The results showed that FZ06 and MS2 contain 491 and 443 unique genes, respectively, including 131 (in FZ06) and 237 (in MS2) genes encoding known functional proteins annotated (Table 2). Based on GO functional annotation, we classified the 439 FZ06 unique genes into 14 classes, among which catalytic activity, binding, and cellular anatomical entity each accounted for about 20% of the genes, followed by metabolic and cellular process (Appendix A). Among the FZ06 unique genes, we also identified five large gene clusters that encode one Stt type II secretion system (Stt-type T2SS), three type IV secretion systems (T4SSs), and one type IV pilus (T4P), respectively (discussed in detail below).

In addition, we compared pathogenicity-related genes of strains FZ06 and MS2, including PCWDEs, T1SS~T6SS secretion systems, flagella, and chemotaxis. Substantial divergence was found between the two banana strains as follows:

*Cell-Wall-Degrading Enzymes* (CWDEs): A sequence search using published CWDE-coding genes in *D. oryzae* EC1 [25] as queries identified more than 30 homologs in both FZ06 and MS2. The two strains shared a common set of genes encoding 15 pectinases (Pnl, PelN, PelW, PelL, PelI, PelA, PelE, PelD, PaeX, PaeY, PemA, PelC, PelB, PelZ, and PelX), three polygalacturonases (PehN, PehK, and PehX), four proteases (PrtX, PrtC, PrtB, and PrtG), eight cellulases (BglA, BgxA, BglB, LfaA, NagZ, CelZ, CelH, and CelY), one rhamnogalacturonate lyase (RhiE1), and one xylanase (XynA). Moreover, two additional copies of polygalacturonases (PehV and PehW) and 6-phospho-beta-glucosidases (BglC and BglD) and one extra copy of rhamnogalacturonate lyase (RhiE2) were uniquely present in the FZ06 genome (Appendix A). Most of these CWDE genes are scattered throughout the genomes, except for the three clusters of *pelAED-paeY-pemA*, *pelCBZ*, and *prtXCBG* (Appendix A).

*Secretion systems*: Both genomes harbor PrtDEF-T1SS, which is responsible for the secretion of proteases, located between the protease-encoding genes *prtXCB* and *prtG*, similar to the gene arrangement in *D. dadantii* 3937 and *D. oryzae* EC1 [25,45,46]. In *Dickeya* spp., two types of T2SS were characterized. One is the Out-T2SS (encoded by *outSBCDEFGHIJKLMO*) conserved in *Dickeya* spp., transporting extracellular proteins, including pectinases and cellulases. The other is the Stt-T2SS, encoded by the *stt* gene cluster (*sttSMLKJIGFED*). The Stt system is present only on the exterior of the outer membrane in *D. dadantii* subsp. *dadantii*, *D. dadantii* subsp. *dieffenbachiae*, *D. chrysanthemi*, and *D. dianthicola* [47,48] but absent in available genomes of *D. fangzhongdai*, *D. oryzae*, *D. zeae*, and *D. solani* [49]. This pattern was confirmed by our finding that both FZ06 and MS2 contain a highly conserved *out* system (encoded by *FZ06_003179*~*FZ06_003193* and *C1O30_RS14365*~*C1O30_RS14430*, respectively). The FZ06 genome also contains the Stt system (encoded by *FZ06_002735*~*FZ06_002745*). No *stt* genes were found in the MS2 genome (Figure 5A).

T3SS, which is encoded by the *dsp*, *hrp*, and *hrc* gene clusters, was recently characterized in MS2 [21]. The core *hrp* and *hrc* gene clusters are highly conserved in the genomes of most *Dickeya* species, including FZ06 and MS2, with some differences between *plcA* and *hecB*. In strain FZ06, *hecA* and *hecB* encoding a T5SS are separated by ~80 genes, and the *FZ06_002475* gene encoding a cupin domain-containing protein is inserted downstream of *hecB*. In strain MS2, two genes encoding hypothetical proteins (*C1O30_11590* and *C1O30_11595*) are located upstream of *plcA* (Figure 5B).

Prediction of T4SS indicated that the FZ06 genome contains two complete VirB-T4SSs (*FZ06_001596~FZ06_001605* and *FZ06_002989~FZ06_002998*, encoding *virB1*~*virB11*) that are almost identical (96.69% nucleotide sequence identity and 100% coverage). Furthermore, an additional Trb-T4SS (*FZ06_000820~FZ06 _000831*, also called P-type T4SS) is also present in the FZ06 genome, whereas all T4SS-encoding genes, except for *virB1* and *virB2*, are missing in MS2 (Figure 5C).

Most *Dickeya* strains contain 17 conserved core genes of T6SS, including *hcp*, *vgrG* (virulence-associated protein G), *impBCF*, and *vasABCDEFGHIJKL*. The T6SS clusters in FZ06 and MS2 are highly conserved on the two sides but differ substantially in the region between *rhsA* and *impB*. Similar to the reported T6SS gene organization in *D. oryzae* EC1, *D. parazeae* Ech586, and *D. dadantii* 3937 [25], two ankyrin-encoding genes are present downstream of *pldA* gene in the FZ06 genome, whereas five copies are present in the MS2 genome (Figure 5D).

*Motility-associated genes*: The MS2 genome contains a set of gene clusters encoding flagellar biosynthesis and chemotaxis proteins with the same gene organization as those of *D. oryzae* strains EC1, DZ2Q, and ZJU1202, as well as *D. zeae* MS1, which was also isolated from banana [25]. A large fatty acid biosynthetic gene cluster ranging from *C1O30_RS13270* to *C1O30_RS13310* is also present in strain MS2, where it is replaced by two genes with functions annotated as a Gfo/Idh/MocA family oxidoreductase and an ATP-grasp domain-containing protein, respectively, in FZ06 (Figure 6A). A conserved *pilMNOPQ* operon encoding a key T4P structural component was also found in the genomes of FZ06 and MS2, whereas an additional pilus assembly gene cluster ranging from *FZ06_000973* to *FZ06_000982* is present in the genome of strain FZ06, encoding PilM2, PilN2, PilO2, PilP2, TadA, GspF, PilX, a lytic transglycosylase domain-containing protein, a hypothetical protein, and PilV (Figure 6B). We searched the genomes of other *Dickeya* strains and found that this gene cluster is only present in FZ06.

*Phytotoxin gene clusters*: In our previous studies, we reported that the rice isolate *D. oryzae* EC1 produces phytotoxic zeamines as a major virulence factor that is encoded by a *zms* gene cluster [25], whereas the banana strain *D. zeae* MS2 produces a novel phytotoxin encoded by an unique gene cluster, *C1030_04995*~*C1030_05185* [36]. Genomic comparison revealed that FZ06 lacks the latter gene cluster but contains the *zms* gene cluster homologous to that of *D. oryzae* EC1 (Appendix A), except for three additional genes designated as *FZ06_001518*~*FZ06_001520* encoding a DUF1272 domain-containing protein, a glycine betaine/L-proline transporter ProP, and a 4’-phosphopantetheinyl transferase superfamily protein.

### 2.5. FZ06 Expressed Lower Levels of CWDE Genes Than MS2

CWDE assays showed that strain FZ06 exhibits significantly lower Pel, Peh, and Prt activities than strain MS2 (Figure 1). However, most CWDE-encoding genes were conserved in both strains, except some missing in the genome of MS2 (Appendix A), contradictory to the phenotypic performance (Figure 1). In our previous study, we discovered that CWDE genes are expressed at significantly lower levels in the monocot-specific *D. zeae* strain JZL7 compared with the *D. zeae* MS2 strain, although they harbor almost the same CWDE repertoire, contributing to the lower virulence of JZL7 [21]. In order to determine whether the lower production of CWDEs by FZ06 is caused by the reduced expression of CWDEs, we measured the expression of all CWDE genes in the two strains. The results showed that most of the 32 shared CWDE genes, as well as three T1SS *prtDEF* genes, were expressed at a lower level in strain FZ06 than in MS2, which was basically consistent with the phenotypic results of extracellular enzymes. In terms of extracellular enzyme categories, the cumulative expression level ratios (FZ06:MS2) of genes encoding Pels, Pehs, Cels, Prts, RhiE1, and XynA were 1.029 (762.170:740.532), 0.769 (131.771:171.281), 0.846 (381.769:451.467), 0.399 (147.405:369.267), 0 (0:2.736), and 0.373 (43.223:115.774), respectively (Figure 7).

## 3. Discussion

*Dickeya* spp. are plant pathogens with a wide host range distributed all over the world. They can infect at least 35% of angiosperm plant orders [50], causing considerable economic losses in global grain, vegetable, and flower production. With an increasing number of *Dickeya* pathogens discovered in recent years, comparative studies have uncovered widespread phenotypic and genomic differences, providing valuable insights into the molecular basis of the pathogenicity and host specificity of *Dickeya* spp. In this study, we reported the complete genome sequence of *D. dadantii* subsp. *paradisiaca* FZ06, a banana soft rot pathogen isolated from the Philippines, characterized its pathogenicity-related phenotypic and genomic features, and compared it with that of *D. zeae* MS2, which was isolated from soft rot banana in China.

During analysis of the relatedness between strain FZ06 and nearby strains, we found that FZ06, S3-1, and NCPPB 2976 were isolated from diseased samples of monocotyledonous hosts (A622-S1-A17 from river water), whereas almost all other *D. dadantii* strains were isolated from dicotyledonous hosts (Appendix A). Notably, the three strains of *D. dadantii* derived from monocotyledonous hosts belong to the same sister branch in the taxonomic status of the evolutionary tree (Figure 4), and they have a closer evolutionary relationship. This finding helps to confirm the identity of the new subspecies of FZ06 (isolated from monocotyledonous hosts with NCPPB 2976 as another subspecies of *D. dadantii*), and their distribution on monocotyledonous host plants may be related to the bacterial evolution and host specificity during plant–pathogen interactions.

Degradation of pectin usually means that the soft rot bacteria has successfully infested the host plant [51]. The infective ability of *D. dadantii* 3937 correlates with its capacity to synthesize and secrete CWDEs. On the other hand, *D. poaceaephila*, isolated from sugarcane in Australia, showed low infestation capacity due to the absence of extracellular enzyme genes [12]. The cumulative expression of CWDE genes is lower in FZ06 (although the repertoire is larger) than in MS2, except for the Pel genes, which might partly explain the lower invasiveness of FZ06 relative to MS2 on most host plants. However, only 5 (PelD, PelE, PemA, PelI, and PelL) of the 15 pectate lyases from *Dickeya* have been reported to be involved in the maceration of hosts. PelD and PelE are the most important enzymes in impregnating plant tissues [52]; PemA, PelI, and PelL were also reported to be involved in virulence on chicory and potato tubers in *D. dadantii* 3937 [53,54,55]. Among them, *pelD* is not expressed, whereas *pemA*, *pelI*, and *pelL* are less expressed in FZ06 (Figure 7). Furthermore, *pelA* and *paeY* have no expression in the *pelAED-paeY-pemA* gene cluster, and *pelN* is more expressed in FZ06 (Figure 7). PelA and PelN exhibit different catalytic properties and differences in gene regulation relative to other pectin lyase isozymes during plant infection, which could allow pectate lyase to play a complementary role in the infestation process, increasing the pathogenicity and pathogenic range of the bacteria [52,56]. PaeY is a pectin acetyl esterase used to remove the acetyl esterification modification of galacturonic acid residues on the pectic polysaccharide backbone and aid in its degradation of host plant tissue pectin [57]. Some Pel and Peh genes are more expressed in FZ06, including *pelCBZ,*
*pelW*, *pelX*, and *paeX* (Figure 7), opposite to the phenotypic results. PelBC cleaves moderately esterified polymers. PelZ regulates *pelB* and is not the main virulence factor of *Dickeya* but associated with host specificity [58]. PelW and PelX exhibit exonuclease activity, sequentially cleaving the oligomers as they pass through the periplasm, as well as into the cytoplasm, to form unsaturated digalacturonic acids, respectively [59]. The differential expression and mutual collaboration of all these pectin lytic enzymes and isozymes might result in specific infestation of bacterial strains to different hosts, which might explain the higher capability of strain FZ06 in infecting individual host plants, such as Chinese cabbage and onion (Appendix A). Among the eight shared Cel genes, *bglA*, *bgxA*, *bglB*, *nagZ*, and *lfaA* genes were more highly expressed in strain MS2, and *celZ*, *celY*, and *celH* genes were more highly expressed in FZ06. The CelZ gene accounted for 95% of the total carboxymethylcellulase activity, and CelY acted synergistically [60]. Therefore, it is reasonable that the high expression of a few important genes in FZ06 resulted in almost equal cellulase phenotypic activities of the two strains. Furthermore, the expression of CWDE genes is regulated by the expression of regulatory factors KdgR, Fis, SlyA, VfmE, PecS, and PecT, and it has also been proposed that the pH-dependent regulatory system of CWDE genes and the expression of isozymes can vary among plants during different periods of infestation [21,61]. It is evident that the secretory and functional roles of CWDE are very complex. The reduced cell-wall-degrading enzymatic activity and virulence of strain FZ06 are, in part, due to its low expression of CWDE-encoding genes.

Because strain FZ06 harbors the whole gene cluster responsible for the biosynthesis of zeamines but did not exhibit any inhibitory activity against *E. coli* DH5α (Figure 3), we inferred that: 1) the antibiotic(s) produced by FZ06 might have different structure(s) from zeamines and therefore might have different microbial inhibitory spectra, 2) FZ06 produces zeamines but could not secrete them to extracellular space, and 3) *zms* genes are not or lowly expressed in FZ06 under the experimental conditions.

Bacterial motility driven by flagella, biofilm, and chemotaxis proteins is associated with the initial infestation, expansion, and signal transduction of the pathogen in the host [62,63]. *D. dadantii* has been shown to exhibit a strong chemotactic response to jasmonic acid (JA), which is produced by wounded plant tissues, thus facilitating the movement of bacterial cells toward plant wounds and promoting systematic plant invasion [51]. FZ06 had significantly lower swimming and swarming motility than MS2, and it remains to be investigated whether this difference is related to functional variation due to the different inserted genes in the flagellin and chemotaxis protein gene clusters (Figure 6). Apart from the flagella-mediated swimming and swarming motility, bacteria twitch by means of the extension and contraction of type IV pilus (T4P) on the bacterial surface [64]. The *pilMNOPQ* operon encodes a key T4P structural component that is essential for pilus biogenesis. Studies on the *pilMNOPQ* operon gene are mostly found in *Pseudomonas aeruginosa* and *Xanthomonas* spp. [65,66]. In addition, it has been shown that *pilQ* mutants of *X**yllela fastidiosa* cannot twitch [67]. Given that there is no difference in twitching movement between FZ06 and MS2, we speculate that the gene cluster (*FZ06 _000973* to *FZ06 _000982*) might not be functional in FZ06. Biofilm formation, which is structurally required for the production of EPS, is essential for bacterial attachment on object/host surfaces and survival under unfavorable environmental conditions [37]. In particular, EPSs form three-dimensional structures that support a variety of biological processes in biofilms, such as bacterial adhesion, colonization, virulence, and protection from extreme environments [68]. The lower EPS production of FZ06 compared to MS2 may have resulted in a weaker colonization capacity on host plants.

Other virulence factors, including bacterial secretion systems, although not directly responsible for the development of soft rot symptoms, are necessary for the full expression of soft rot disease. For example, antioxidant enzymes can eliminate the reactive oxygen species (ROS) produced by host plants, allowing *Dickeya* to break through the first line of plant defense [69,70]. T3SS can trigger HR on host and non-host plants and has been demonstrated to be involved in the virulence of *D. dadantii* 3937 and *D. zeae* MS2 [71,72]. Our recent study revealed that T3SS plays an important role in defining the host range and virulence of the pathogen [21]. T4SS is involved in the splice transfer of bacterial DNA and can deliver effector proteins (virulence factors) directly to the host cell [73], contributing to host adaptation and colonization of soft rot bacteria [74]. T4SS-deficient strains are common in *Pectobacterium* and *Dickeya* spp., and the copy numbers of T4SS vary between strains [73,75]. T6SS plays a role in bacterial pathogenicity and host adaptation [74,76]. Most of the secretion systems (T1SS, Out-T2SS, T3SS, T5SS, and T6SS) are conserved in both FZ06 and MS2, except for the additional Stt-T2SS and three additional T4SS systems in FZ06. The implication of these genomic differences with respect to the virulence of FZ06 remains unclear and requires further investigation.

In summary, the genomic and phenotypic data of FZ06 reported in this study add to our knowledge base of *Dickeya* spp. The comparison of FZ06 and MS2, two *Dickeya* pathogens isolated from the same host, will help to understand the genetic basis of pathogenicity and host specificity of *Dickeya* interspecific strains and will contribute to the development of future control strategies for emerging pathogens.

## 4. Materials and Methods

### 4.1. Bacterial Strains and Growth Conditions

*D. zeae* MS2 isolated from banana in Guangdong Province, China [5,36] and *Dickeya* strain FZ06 isolated from Philippine banana were used in this study. The strains were grown in LB medium (typtone 10.0 g/L, yeast extract powder 5.0 g/L, NaCl 10.0 g/L, pH 7.0, 1.5 g/L agar added in solid medium) at 28 °C [77].

### 4.2. Measurement of Bacterial Growth Curves

Strains FZ06 and MS2 were incubated in LB medium until an OD_600_ of 1.5 was reached, diluted into fresh LB medium at a ratio of 1:100, mixed thoroughly, transferred (500 μL) to 2.0 mL tubes, incubated at 28 °C, and shaken at 200 r/min. The cell density values were measured every 4 h. LB medium was used as a blank control. The experiment was repeated in triplicate.

### 4.3. Pathogenicity Assay against Monocotyledonous and Dicotyledonous Plants

Strains FZ06 and MS2 were incubated in LB liquid medium until an OD_600_ of 1.5 was reached. Cucumber (*Cucumis sativus*), wax gourd (*Benincasa hispida*), Chinese cabbage (*Brassica rapa* subsp. *pekinensis*), radish (*Raphanus sativus*), carrot (*Daucus carota*), potato (*Solanum lycopersicum*), onion (*Allium cepa*), ginger (*Zingiber officinale*), and taro (*Colocasia esculenta*) were sliced (5 mm in thickness) and inoculated with 2 μl of bacterial cultures; tomato (*Lycopersicon esculentum*) and eggplant (*Solanum melongena*) fruit were inoculated with 100 μL of bacterial cultures; and banana (*Musa* AAA and *Musa* ABB) seedlings were inoculated with 200 μL of bacterial cultures. The same volume of LB medium was inoculated as a blank control. Each treatment was repeated three times. The inoculated plants were maintained in a greenhouse (28 ± 2 °C, 75% ± 15% relative humidity, and 12 h alternating light and dark cycles) until symptoms appeared. The area of the lesions was measured using Image J 1.52a software [78]. Banana seedlings were observed after 15 days, and the diseased tissue was weighed [79].

### 4.4. Measurement of Plant-Cell-Wall-Degrading Enzymatic (PCWDE) Activities

Strains FZ06 and MS2 were incubated in LB medium until an OD_600_ of 1.5 was reached. The activities of plant-cell-wall-degrading enzymes were measured using previously described medium formulations [80,81]. The assay medium plates were punched (5 mm in diameter), and 20 µL of bacterial cultures was added into the wells and incubated at 28 °C. Pectate lyase (Pel) and polygalacturonase (Peh) plates were treated with 1 M HCl for 15 min after 12 h. Cellulase (Cel) plates were stained with 0.1% (*w*/*v*) Congo red for 15 min after 12 h and decolorized with 1 M NaCl. Protease (Prt) plates were observed after 24 h without any treatment. The diameter of transparent halos was measured. The experiment was repeated in triplicate.

### 4.5. Phytotoxin Assay

Strains FZ06 and MS2 were incubated in LB medium to an OD_600_ of 1.5. To prepare the phytotoxin assay plate [40], 20 mL of melted LB agar medium was poured into a square plastic plate (10 × 10 cm); after solidification, 15 mL of 1% agarose cooled to 50 °C was inoculated with *E. coli* DH5α (10^8^ CFU) and laid on the medium. The plate was punched (5 mm in diameter), and 20 µL of FZ06 or MS2 culture was added into each well and incubated at 28 °C for 16 h. The diameter of antagonistic circles was measured. The experiment was repeated in triplicate.

### 4.6. Measurement of Extrapolysaccharide (EPS) Production

The EPS yields were determined as previously described [5]. Strains FZ06 and MS2 were incubated in LB medium until an OD_600_ of 1.5 was reached and inoculated into 300 mL of LB fresh medium at a ratio of 1:100 for culture with shaking at 200 rpm for 12 h. Bacterial cultures were centrifuged at 5000 rpm for 40 min, and the supernatants were collected and volume-recorded. Twice the volume of anhydrous ethanol was added into the supernatants and mixed thoroughly, then precipitated overnight at 4 °C. The supernatants were discarded, and the precipitate was dried and weighed at 55 °C. The content of EPS per unit volume was calculated. The experiment was repeated in triplicate.

### 4.7. Measurement of Cell Motility

To test the motility of strains FZ06 and MS2, swimming medium (0.5% peptone, 0.5% NaCl, and 0.25% agar), swarming medium (1% tryptone, 0.5% NaCl, and 0.4% agarose), and twitching medium (0.5% yeast extract, 1% tryptone, 1% NaCl, and 1% agarose) plates were prepared as previously described [40]. The strains were incubated in LB liquid medium until an OD_600_ of 1.0 was reached. Specifically, 1 µL of the bacterial culture was added to the inside of the swimming medium, the surface of the swarming medium, and the bottom of the twitching medium, respectively, and incubated at 28 °C for 12, 8, and 24 h, respectively. The diameter of bacterial motility was measured and collated with Image J 1.52a. The experiment was repeated in triplicate.

### 4.8. Genomic Sequencing, Assembly, and Annotation

A single colony of strain FZ06 was incubated in LB medium until an OD_600_ of 1.0 was reached, 4 mL of which was used for genomic DNA extraction using an EasyPure bacteria genomic DNA kit (EPICENTRE Biotechnologies, Madison). Then, DNA quality was checked by agarose gel electrophoresis, quantified using a Qubit 2.0 fluorometer (Thermo Scientific, Waltham, MA, USA), and sent to Novogene (Tianjin, China) for whole-genome sequencing using both nanopore PromethION and Illumina NovaSeq platforms. A de novo assembly of nanopore long-read sequencing data was performed using Flye v2.7 [82]. The assembly was subsequently polished by BGISEQ short-read sequencing data using Pilon v1.24 [83]. The sequencing data and genome assembly were deposited in the NCBI database under GenBank accession no. CP094943.1.

A circular visualization of FZ06 genome characteristics was created by Circos [84]. The FZ06 genome was annotated using the Prokaryotic Genome Annotation Pipeline (PGAP, 2021-01-11.build5132) [85]. TXSScan was used to annotate secretion systems [86].

### 4.9. Phylogenetic and Genomic Analyses

Pairwise average nucleotide identity (ANI) values between FZ06 and all 167 *Dickeya* genomes available in the NCBI RefSeq database (Appendix A; accessed on 16 March 2022) were calculated using fastANI v1.3 [44]. Digital DNA-DNA hybridization (dDDH) values between FZ06 and *D. dadantii* strains that share the highest ANI values with FZ06 (over 95%) were then determined using the GGDC server (http://ggdc.dsmz.de/). Orthologous gene groups were constructed using OrthoFinder v2.5.4 [87] from annotated proteins in the genomes of FZ06, 16 *D. dadantii* strains, and 11 type strains of other *Dickeya* species with default settings, except that the sensitivity of diamond search was set to “ultrasensitive”. The resultant “tsv” file was used in downstream analyses, such as the identification of unique genes or orthologs of previously reported SRP genes encoding secretion systems and PCWDEs [21,25,88,89]. Phylogenetic analysis of FZ06, all 16 *D. dadantii* strains, and 11 type strains of other *Dickeya* species was performed based on a set of 1264 single-copy orthologous genes. In detail, single-copy genes were aligned using MAFFT v7.490 (with the “E-INS-i” iterative refinement method) [90], trimmed using trimAl v1.4 (with the “gappyout” option) [91], and concatenated into one supermatrix. Phylogenetic inference was performed using IQ-TREE v2.1.2 with an automatic selection of the best-fitting evolutionary model [92], and the reliabilities of internal branches were estimated by ultrafast bootstrap analysis with 1000 replicates.

### 4.10. RNA Purification and Reverse Transcription-PCR (RT-PCR) Analysis

Single colonies of strains MS2 and FZ06 were cultured to an OD_600_ of approximately 1.0 for RNA extraction. RNA was extracted with an SV total RNA isolated system kit (Promega, Madison, WI, USA) and further purified with an RNA clean kit (Qiagen, Hilden, Germany). DNA contamination was eliminated with DNase I. RNA purity was checked by gel electrophoresis, and RNA quality was determined by NanoDrop 2000c (Thermo Fisher Science, Waltham, MA, USA). 

For RT-PCR analysis, 1 µg of RNA was reverse-transcribed to generate template cDNA using a HiScriptII Q RT SuperMix kit (Vazyme Biotech Co., Nanjing, China). The expression of the internal reference gene 16S rRNA (*FZ06-000214*) was chosen to equilibrate the concentration of cDNA in strains MS2 and FZ06. The primers used for PCR amplification were newly synthesized in this experiment and are listed in Appendix A. The RT-PCR was performed using 2 × Rapid Taq Master Mix (Vazyme Biotech Co., Nanjing, China) with the following cycling patterns: 1 cycle at 95 °C for 3 min; followed by 30 cycles at 95 °C for 15 s, 60 °C for 15 s, and 72 °C for 15 s; 1 cycle at 72 °C for 5 min; and holding at 16 °C. Owing to the multiple copies of 16S rRNA in the MS2 and FZ06 genomes, to observe slight changes in the PCR products, the cycles were set as 16 for 16S rRNA gene amplification. The expression of each gene was determined using Image Lab software (Bio-Rad, Hercules, CA, USA) by measuring the signal intensities of the bands [21]. The experiment was repeated in triplicate.

## Figures and Tables

**Figure 1 ijms-23-12758-f001:**
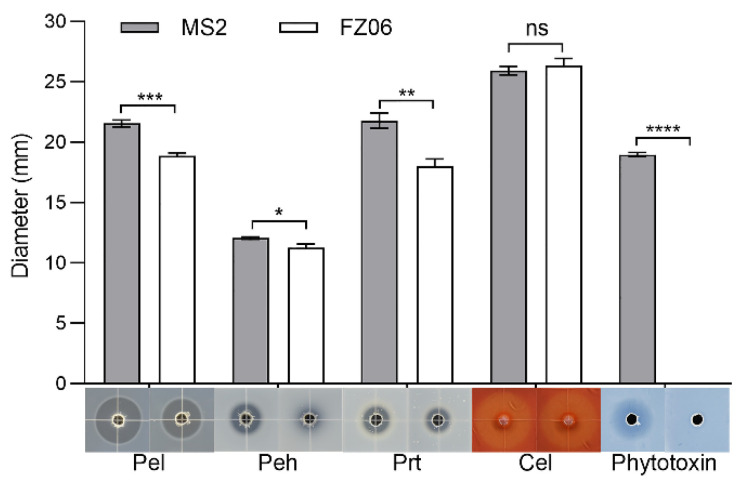
Plant-cell-wall-degrading enzymes (PCWDEs) and phytotoxins produced by strains FZ06 and MS2. Twenty microliters of MS2 or FZ06 culture (OD_600_ of 1.5 in LB medium) was added into each well (5 mm in diameter) of the assay plates and incubated at 28 °C until photographed. The diameter of transparent and antagonistic halos was measured. GraphPad Prism 8.4.1 was used to perform unpaired two-tailed *t*-tests, and the data of strain FZ06 were compared with those of strain MS2. The data present the means of three replicates, and error bars represent the standard deviation. “ns” indicates not significant, * indicates *p* < 0.05, ** indicates *p* < 0.01, *** indicates *p* < 0.001, and **** indicates *p* < 0.0001.

**Figure 2 ijms-23-12758-f002:**
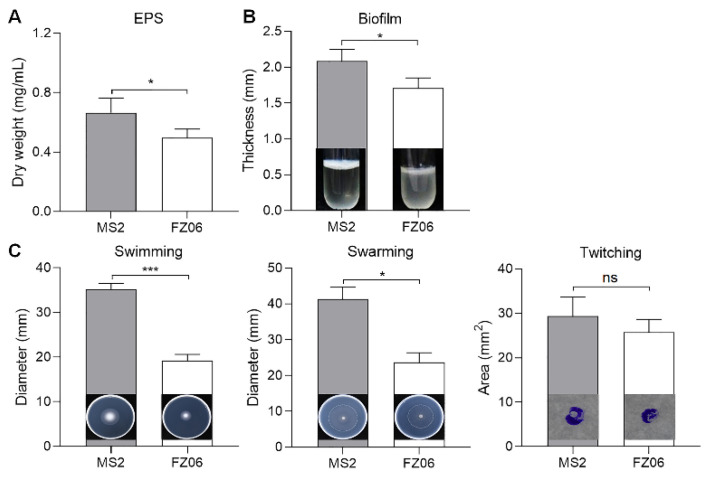
Measurement of EPS (**A**), biofilm formation (**B**), and cell motility (**C**) of strains FZ06 and MS2. GraphPad Prism 8.4.1 was used to perform unpaired two-tailed *t*-tests, and the data of strain FZ06 were compared with those of strain MS2. The data represent the means of three replicates, and error bars represent the standard deviation. “ns” indicates not significant, * indicates *p* < 0.05, and *** indicates *p* < 0.001.

**Figure 3 ijms-23-12758-f003:**
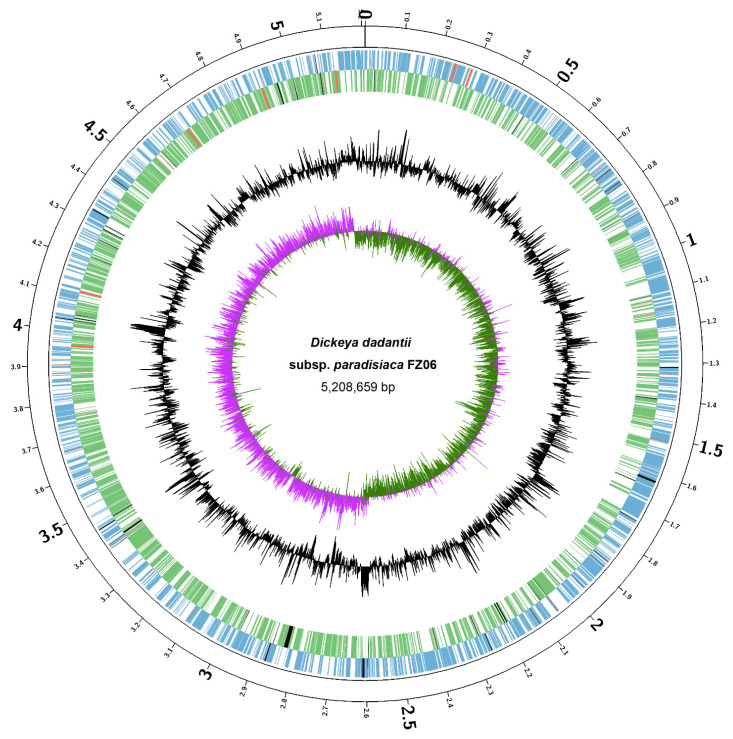
Circular chromosome map of *D. dadantii* subsp. *paradisiaca* FZ06. The circles (from outside to inside) represent features of the positive strand, showing coding sequence (CDS) (blue), rRNA (red), and pseudogenes (black); features of the negative strand, showing CDS (green), rRNA (red), and pseudogenes (black); GC content; and GC-skew (pink and green indicate values higher and lower than the mean value, respectively).

**Figure 4 ijms-23-12758-f004:**
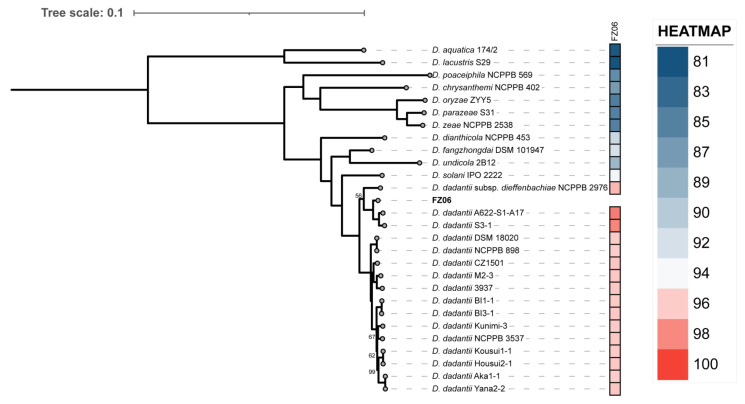
Phylogenetic tree based on 1264 bacterial conserved orthologous single-copy genes of strain FZ06, 16 *D. dadantii,* and 11 type strains of other *Dickeya* spp. Single-ANI analysis was used by means of fastANI v1.3 [44]. The closest strain of FZ06 is *D. dadantii* A622-S1-A17, with a 98.4955% ANI value.

**Figure 5 ijms-23-12758-f005:**
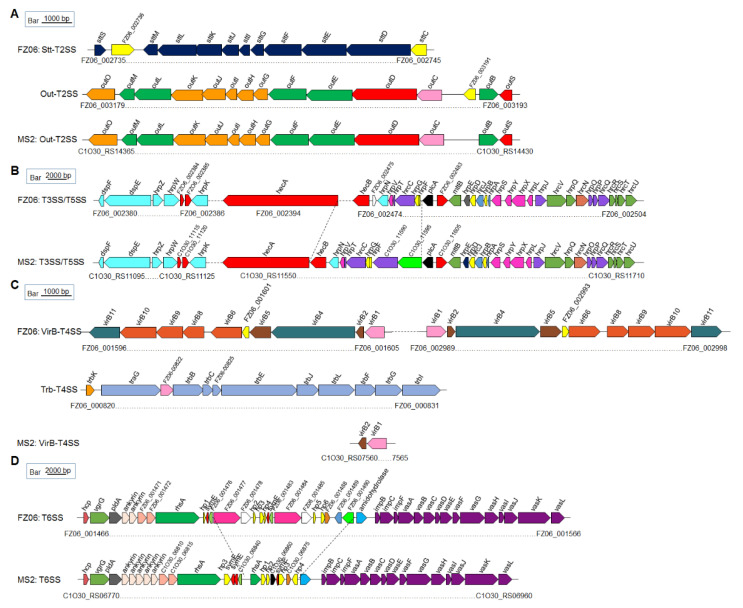
Genetic organization of the T2SS (**A**), T3SS and T5SS (**B**), T4SS (**C**), and T6SS (**D**) clusters in strains FZ06 and MS2. ORFs with the same color in each panel represent the same biological function.

**Figure 6 ijms-23-12758-f006:**
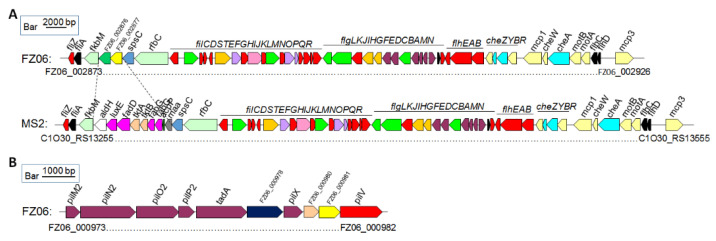
Genetic organization of the motility and chemotaxis gene clusters in strains FZ06 and MS2. (**A**). Flagellar and chemotaxis gene clusters in the genomes of FZ06 and MS2; (**B**). The pilus assembly protein clusters in the genomes of FZ06 and MS2.

**Figure 7 ijms-23-12758-f007:**
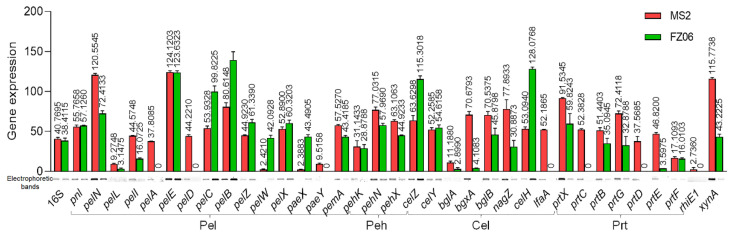
RT-PCR of CWDE genes of strains MS2 and FZ06 (OD_600_ of 1.0). The reference gene 16S rDNA was used to equilibrate the concentration of cDNA samples. The expression of genes was determined by measuring the signal intensity of the bands (under the x axis) using Image Lab software (Bio-Rad, Hercules, CA, USA). Experiments were repeated in triplicate, and the mean data above the bars indicate the signal intensity of RT-PCR bands.

**Table 1 ijms-23-12758-t001:** Lesion sizes or weights on 13 dicotyledonous and monocotyledonous plants caused by strains FZ06 and MS2.

Inoculated Plant			Inoculation Amount, Time	Unit of Diseased Tissue	Inoculated Strain ^a^	
Class	Species	Organ			MS2	FZ06
Dicots	*Cucumis sativus*	Fruit	2 μL, 12 h	mm^2^	346.40 ± 11.37 aA	202.19 ± 18.77 bB
	*Benincasa hispida*	Fruit	2 μL, 12 h	mm^2^	1157.58 ± 28.99 bB	1345.47 ± 5.92 aA
	*Brassica**rapa* subsp. *pekinensis*	Tuber	2 μL, 12 h	mm^2^	20.09 ± 3.102b B	154.39 ± 5.832 aA
	*Raphanus sativus*	Tuber	2 μL, 12 h	mm^2^	415.70 ± 11.14 aA	435.64 ± 15.86 aA
	*Daucus carota*	Tuber	2 μL, 12 h	mm^2^	566.42 ± 18.80 aA	589.61 ± 19.63 aA
	*Solanum lycopersicum*	Tuber	2 μL, 12 h	mm^2^	360.69 ± 20.31 aA	120.49 ± 8.61 bB
	*Lycopersicon esculentum*	Fruit	100 μL, 12 h	mm^2^	108.00 ± 4.563 aA	97.14 ± 2.544 aA
	*Solanum melongena*	Fruit	100 μL, 12 h	mm^2^	443.00 ± 6.02 aA	429.79 ± 8.469 aA
Monocots	*Allium cepa*	Bulb	2 μL, 12 h	mm^2^	9.73 ± 2.17 bB	60.68 ± 4.08 aA
	*Zingiber officinale*	Tuber	2 μL, 12 h	mm^2^	215.15 ± 7.05 aA	173.01 ± 12.15 aA
	*Colocasia esculenta*	Tuber	2 μL, 24 h	mm^2^	153.91 ± 33.1 aA	160.34 ± 10.76 aA
	*Musa*, ABB	Stem	200 μL, 14 d	g	5.83 ± 0.91 aA	0.80 ± 0.12 bB
	*Musa*, AAA	Stem	200 μL, 14 d	g	7.18 ± 0.03 aA	4.00 ± 0.70 bA

^a^ Data were statistically analyzed using ANOVA, and significantly different values (*p* < 0.05) are indicated by different letters.

**Table 2 ijms-23-12758-t002:** Genomic features of strains FZ06 and MS2.

Feature	FZ06	MS2
GenBank accession number	CP094943.1	CP025799.1
Size (bp)	5,208,659	4,740,052
GC content (%)	56.42	53.45
Genes	4582	4221
CDS	4476	4118
RNA genes	106	103
rRNA	22	22
tRNA	75	75
ncRNA	9	6
Pseudogenes	80	50
Proteins	4396	4068
Predicted unique genes	491	443
Function assigned	131	237
Function unknown	360	206

## Data Availability

The data presented in this study are openly available in NCBI database at https://www.ncbi.nlm.nih.gov/nuccore/CP094943.1 accessed on 16 March 2022, reference number CP094943.1.

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
