# Peer review of "Comparative Pathogenomic Analysis of Two Banana Pathogenic Dickeya Strains Isolated from China and the Philippines"

_ijms, 2022, doi:10.3390/ijms232112758_

Round 1

Reviewer 1 Report

The manuscript requires significant revisions and cannot be adopted in its current form.   

The authors of the article submitted for review should not put forward a proposal for the delineation of a new subspecies of Dickeya dadantii subsp. paradisiaca. Describing new taxa requires the fulfilment of specific standards. Authors should provide a very precise description of characteristics based on the results of a comprehensive analysis of phenotypic and genetic traits and not just genomic analyses. In addition to biochemical properties, chemotaxonomic analyses should also be performed. It is necessary to indicate the features based on which a new taxon can be distinguished from others.  

Moreover, a new and separate taxon is not separated for only one strain, which is the case in this study.   

It is necessary to deposit the strain in at least two international collections. Furthermore, the authors should justify the proposed name and indicate the type strain for the new taxon.   

The authors did not provide such data, so in my opinion, authors should remove the proposal for the identification of a new subspecies and remain only with a comparative analysis of two Dickeya strains isolated from bananas.  

The organisation of the text in the results section should be tidied up. For example, at the begging of this paragraph, there is a description of the genome sequence and phylogenomic analyses followed by a description of phenotypic properties such as virulence or activity of PCWD enzymes, and then again, a description of the genomic analyses is continued.  

The description of unique genes is too laconic and requires more details.  

Genome sequences are not available in the Genbank and to the reviewer.  

The correct accession numbers for the genome described are not provided.  

Conditions for the RT-PCR reaction are not given, nor are appropriate references provided in which this would be described. In addition, if the primers are newly designed, there is no description of PCR standardisation.  

Specific comments 

Introduction:  

Line 31  

There is an error in the species name; the last letter is missing; it should be Dickeya aquatica  

In lines 170 and 171, a typo should be corrected, and Treptone should be replaced by Tryptone.  

Methods:  

Line 99  

The PRJNA821986 number given is the project number and not the accession number in GenBank. Please correct to the correct one.  

Line 189   

There is no information available as to whether the RT PCR primers used, as listed in Table S4, were designed in this study or taken from literature. If from the literature, appropriate information should be given (in the table) for each primer pair.  

If these are new primers, then information on RT-PCR reaction conditions is also missing.  

Results: 

Table 1 -  

The species name for strain FZ06 is missing from the title of Table 1. Therefore, either the authors list the species names for both strains or none.  

It would be useful also to provide accession numbers for both genomes in the table.  

Line 207.  

Please specify whether the 16S gene was sequenced or extracted from the genomic sequence of strain FZ06? If this gene was sequenced separately, please provide the accession number.  

Figure 2 - line 229  

Clarify the title based on what the tree presented in figure 2 was generated. Were these orthologous gene sequences?  

Table 2   

Why were the DDH values not counted for the 11 genomes of Dickey species other than D. dadantii?  

Line 255  

Remove the extra dot before (Fig.3)  

In the same lane …, whereas strain MS2 could? Shouldn't it be did instead of could?  

Lines 255-257  

This sentence is incomprehensible. Was something accidentally deleted?  

Figures 3 I 4  

The position of the indications of statistical significance (asterisks * and ns) in figures 3 and 4 is unfortunate because it can mean both the significance of differences between replicates for a given strain and between strains.  

Lines 281- 286  

The description of unique genes is too laconic and requires more details. Figure S3 and Table S3 do not contribute information about what the unique genes encode or what processes they are involved in. Neither the COG class name nor the GO ID alone contributes much. Supplementary tables can be supplemented with annotations of individual genes, and only the most important ones can be mentioned in the text of the manuscript.  

The paragraph concerning RT-PCR 370-390  

Why was only 1 reference gene for RT-PCR reaction used?  

Why was the 16S rRNA gene chosen? It is a multi-copy gene, and most often, several housekeeping genes are used.   

Figure 7 is not legible; the fonts on the graph are too small.  

Discussion: 

In the introduction and results paragraphs, the authors use the Latin names of the plants used in the pathogenicity tests, but the English names are used in the discussion. The way of naming should be standardised throughout the manuscript.  

Lines 401-408  

This part of the discussion is a repetition of the results.  

Some sections of this part of the manuscript do not seem essential in the discussion of the results, e.g. lines 409-415. 

Author Response

The manuscript requires significant revisions and cannot be adopted in its current form.   

The authors of the article submitted for review should not put forward a proposal for the delineation of a new subspecies of Dickeya dadantii subsp. paradisiaca. Describing new taxa requires the fulfilment of specific standards. Authors should provide a very precise description of characteristics based on the results of a comprehensive analysis of phenotypic and genetic traits and not just genomic analyses. In addition to biochemical properties, chemotaxonomic analyses should also be performed. It is necessary to indicate the features based on which a new taxon can be distinguished from others.  

Moreover, a new and separate taxon is not separated for only one strain, which is the case in this study.   

It is necessary to deposit the strain in at least two international collections. Furthermore, the authors should justify the proposed name and indicate the type strain for the new taxon.   

The authors did not provide such data, so in my opinion, authors should remove the proposal for the identification of a new subspecies and remain only with a comparative analysis of two Dickeya strains isolated from bananas.  

>> Thanks for the comment, with which we respectfully disagree. International Journal of Systematic and Evolutionary Microbiology is the main forum for the publication of novel microbial taxa, and in 2021 they presented a proposal for the description of new species as follows "Chemotaxonomic methods played an important role in the development of the polyphasic approach to classification of Archaea and Bacteria. However, we here argue that routine application of these methods is unnecessary in an era when genomic data are available and sufficient for species delineation. Thus, authors who choose not to utilize such methods should not be forced to do so during the peer review and editorial handling of manuscripts describing novel species. Instead, we argue that chemotaxonomy will thrive if improved analytical methods are introduced and deployed, primarily by specialist laboratories, in studies at taxonomic levels above the characterisation of novel species". We have attached some of the original articles and links below: https://www.microbiologyresearch.org/content/journal/ijsem/10.1099/ijsem.0.005127; https://www.microbiologyresearch.org/content/journal/ijsem?page=about-journal.

We hope that the work we have done on the phenotypic and genomic analysis on FZ06 will support our proposal for a new subspecies of D. dadantii. If the reviewers do not agree with this, we are willing to add relevant experiments such as fatty acid analysis, or delete the proposal of a new subspecies, only can the manuscript be accepted for publication.

Regarding the comment that "a new and separate taxon is not separated for only one strain ", we mentioned in the acknowledgement section that our strain FZ06 was mailed from Jianping Yi of Shanghai Entry-Exit Inspection and Quarantine Bureau. Unfortunately, we only got this one isolate, and we are sorry that it is impossible for us to obtain other isolates due to sample discard.

The organisation of the text in the results section should be tidied up. For example, at the begging of this paragraph, there is a description of the genome sequence and phylogenomic analyses followed by a description of phenotypic properties such as virulence or activity of PCWD enzymes, and then again, a description of the genomic analyses is continued.  

 >> Thanks for the comments. We have rearranged the order of the sections of Results and Methods & Materials. We firstly posted the phenotypic properties of the two strains, and then presented their genomic characteristics.

The description of unique genes is too laconic and requires more details.  

 >> Thanks for the comment. We have added an additional Table S3 to provide the annotation information of each unique gene in FZ06.

Genome sequences are not available in the Genbank and to the reviewer.  The correct accession numbers for the genome described are not provided.  

 >> Thanks for the comment. We have replaced the Genbank accession no. by BioProject accession no. since the genome of FZ06 has not been released by NCBI. The link of FZ06 information is: https://www.ncbi.nlm.nih.gov/bioproject/?term=PRJNA821986.

Conditions for the RT-PCR reaction are not given, nor are appropriate references provided in which this would be described. In addition, if the primers are newly designed, there is no description of PCR standardisation.  

>> Thank you for your comments. The RT-PCR primers designed in this study have been listed in Table S6, and information on RT-PCR reaction conditions has been added as suggested. Since this is RT-PCR, not RT-qPCR, we did not performed PCR standardization.

Specific comments  

Introduction:  

Line 31  

There is an error in the species name; the last letter is missing; it should be Dickeya aquatic 

>> Modified as suggested.

In lines 170 and 171, a typo should be corrected, and Treptone should be replaced by Tryptone.  

>> Corrected as suggested.

Methods:  

Line 99  

The PRJNA821986 number given is the project number and not the accession number in GenBank. Please correct to the correct one.  

>> Thanks for the suggestion. We have replaced the Genbank accession no. by BioProject accession no.

Line 189   

There is no information available as to whether the RT PCR primers used, as listed in Table S4, were designed in this study or taken from literature. If from the literature, appropriate information should be given (in the table) for each primer pair.  

If these are new primers, then information on RT-PCR reaction conditions is also missing.  

>> Thank you for your comments. The RT-PCR primers listed in Table S6 were newly designed in this study and the information on RT-PCR reaction conditions has been added to the methods.

Results: 

Table 1 -  

The species name for strain FZ06 is missing from the title of Table 1. Therefore, either the authors list the species names for both strains or none.  

It would be useful also to provide accession numbers for both genomes in the table.  

>> Thanks for your suggestion. We have deleted the species names of strain MS2 and added the BioProject nos. of the two genomes in the table.

Line 207.  

Please specify whether the 16S gene was sequenced or extracted from the genomic sequence of strain FZ06? If this gene was sequenced separately, please provide the accession number.  

>> Thank you for your comment. The 16S gene sequence in this study was extracted from the whole genome sequence of strain FZ06, and we have provided the gene accession no. in the section of Materials and Methods.

Figure 2 - line 229  

Clarify the title based on what the tree presented in figure 2 was generated. Were these orthologous gene sequences?  

>> Figure 2 was generated based on 1264 bacterial conserved single copy genes and we have corrected the captions. These genes are orthologous in different Dickeya strains.

Table 2   

Why were the DDH values not counted for the 11 genomes of Dickey species other than D. dadantii?  

>> Thank you for your comments. We first calculated the ANI values of the FZ06 genome with other genomes of the genus Dickeya, as shown in Figure 4 and Table S1, and found that FZ06 should be classified under the species of D. dadantii. Secondly, we wanted to know the genomic differences between FZ06 and other D. dadantii species, so we further calculated their DDH values.

Line 255  

Remove the extra dot before (Fig.3)  

In the same line …, whereas strain MS2 could? Shouldn't it be did instead of could?  

>> Corrected as suggested.

Lines 255-257  

This sentence is incomprehensible. Was something accidentally deleted?  

>> You’re so intelligent! Yes, we accidentally deleted a sentence “EPS is another virulence determinant in Dickeya pathogens.” Here, and we have added before the second dot.

Figures 3 I 4  

The position of the indications of statistical significance (asterisks * and ns) in figures 3 and 4 is unfortunate because it can mean both the significance of differences between replicates for a given strain and between strains.  

 >> We’re very grateful for this suggestion. We have put the statistical significance (asterisks * and ns) between different strains in the two figures.

Lines 281- 286  

The description of unique genes is too laconic and requires more details. Figure S3 and Table S3 do not contribute information about what the unique genes encode or what processes they are involved in. Neither the COG class name nor the GO ID alone contributes much. Supplementary tables can be supplemented with annotations of individual genes, and only the most important ones can be mentioned in the text of the manuscript.  

>> Thank you for your suggestions. We have accepted the reviewer's suggestion to add the functional annotations of individual unique genes in the FZ06 genome in Table S3. Table S4 and Figure S3 serve as a categorical summary of the information encoded by or processes involved in these unique genes. Important unique genes we distinguished by the function of the gene cluster they encode, have also been described separately in the later sections.  

The paragraph concerning RT-PCR 370-390  

Why was only 1 reference gene for RT-PCR reaction used?  

Why was the 16S rRNA gene chosen? It is a multi-copy gene, and most often, several housekeeping genes are used.   

>> Regarding your question, we are happy to explain. At the beginning, we tried to use several housekeeping genes including infB, atpD, rpoB and 16S rRNA to adjust the quantity of MS2 and FZ06 templates. We found that the expression of infB, atpD and rpoB is differential in the two strains, even when we quantified the FZ06 and MS2 RNA templates in the same amounts. However, similar expression level could be obtained from both templates by using 16S rDNA primers to amplify the bands with the PCR cycle of 15. Thus, we chose 16S rDNA as the internal reference gene to equilibrate the concentration of cDNA in strains MS2 and FZ06.

Figure 7 is not legible; the fonts on the graph are too small.  

>> We apologize for this and have enlarged the fonts in the figure.

Discussion: 

In the introduction and results paragraphs, the authors use the Latin names of the plants used in the pathogenicity tests, but the English names are used in the discussion. The way of naming should be standardised throughout the manuscript.  

>> Thank you for your comments. This part is duplicated with the results, and we have deleted it as suggested below.

Lines 401-408  

This part of the discussion is a repetition of the results.  

>> Thank you for your comment. We have removed this duplicate description.

Some sections of this part of the manuscript do not seem essential in the discussion of the results, e.g. lines 409-415. 

>> Thank you for your suggestion. We have significantly trimmed these sentences.

Reviewer 2 Report

1 Overall evaluation:

I believe that this research has sufficient merit to be published. The biological problem and the organism under analysis are important and the results provide new information useful for understanding the species richness of the genus Dickeya, together with important differences in terms of virulence between its members. The methodology was adequate and the authors presented well-supported conclusions. I have two main concerns that I would like the author of the article to address in the manuscript:

1)  Why, if the recognition of a new subspecies of D. dadantii is proposed, all comparisons are made with D. zeae and not with the first. It was very interesting to see the results related to the comparison of both species, and now we know well how the new subspecies differ from D. zeae but not from D. dadantii.

2) The proposal of a new subspecies deserves that the authors spend some time justifying it. Or is it enough to mention only in the Results section that a specific ANI value is sufficient to taxonomically reach such a conclusion? If that's the case, the article loses most of its merits. Present ALL the criteria and facts that you think justify the taxonomic creation of this subspecies.

2 The manuscript:

The article is easy to read, but sometimes one finds shocking sentences that all derive from inaccurate use of language. Details are provided below. However, after making all the suggested corrections, ask a person whose level of English is native to proofread the manuscript.

Specific comments:

- Line 11: ..."FZ06 belongs to D. dadantii"... belongs????

- Line 20: delete "Findings in" and rephrase "...clarified the genetic differentiation of Dickeya interspecific strains associated with pathogenicity...." because it makes no logical sense in the way it is written.

- Line 42: Change "and brought" to "that brought".

-Line 46: Current name Streptocarpus ionanthus.

-Line 56: The accepted name is Canna indica.

-Line 75: "belongs to D. dadantii".... belongs????

-Line 78: change "banana" to "bananas".

- Lines 78/79: Change "of the two strains" to "of the two strains MS2 and FZ06".

- Line 86: Reference? LB is a well-known media and recipe but it was devised and created by somebody...

- Line 104: change the title of the protocol to something like "Phylogenetic and genomic analyses" since the current title is a result.

- lines 118/123: References for all these methods and programs are missing.  

- Line 132: Winter squash is Cucurbita maxima. Benicasa hispida is the scientific name of what is known as the wax gourd. So, here... Which one was it?

-Line 132: The correct designation of Chinese cabbage is Brassica rapa subsp. pekinensis.

- Line 135: the scientific name of tomato is Solanum lycopersicum.

- Line 138: this was NOT a negative control but a 'Blank'.

- Line 141: Please rephrase "pictured till diseased symptoms were observed". What's that supposed to mean???

- LInes 145/151: Please provide references or state that all those methods were developed in your lab.

- Lines 153/159:  I guess there are references (at least one) for all these media and methods... aren't they?

- Line 166: Explain "supernatants were discarded, dried at 55°C and weighed" better (i.e., you dried what was discarded?) and provide a reference for the method.

- Line 170: Change "Trpetone" to "Tryptone".

- Lines 169/177: Please provide references.

- Lines 199/200: Change "protein-coding genes, 75 tRNAs, 9 ncRNAs, and 22 rRNAs..." to "protein-coding, 75 tRNAs, 9 ncRNAs, and 22 rRNAs genes..."

- Line 206: Please check the correct use of plurals (e.g., pseudogenes, not pseudogene, and so on) in Table 1

- Line 217: I prefer you use "relatedness analysis" instead of phylogeny (which is not).

- Lines 224/227: If this is really relevant please move it to the Discussion section and explain why it is important.

- Line 229: I would suggest changing the title of this Figure.

- Line 232: Please consider sending Table 2 to Supplementary material.

- Lines 243-246: Please check the correct use of taxonomic names taking into consideration the observations made before.

- Line 247: Please change the title of this Table to something more accurate and informative. For example, What are those values? Lesions sizes?

- Lines 251-252: Please rephrase, it is confusing. Make emphasis on the virulence differences among strains; in the way the sentence is written, it seems that you point to differences among the virulence factors themselves.

- Line 265: Please change "till pictures captured" to "until photographed".

- Lines 267 and 273: change "performed" to "perform".

- Line 394: Please rephrase "crop, food, vegetable, and flower industries." What is a crop industry?, for example.

- Line 430: What is "Although the virulence of PelA and PelN infestation was relatively low" suppose to mean? To me, this sentence does not make sense at any level.

- Line 458: change "part due" to "in part due".

- Line 465: Change "film (pellicle)" to "biofilm".

- Line 478: This is confusing and misleading: you are talking about Xanthomonas spp and then jump to mention X. fastidiosa. Are you referring to Xyllela fastidiosa? Because that's a different bacterial genus. Furthermore, the reference provided is about Xanthomonas oryzae pv. oryzae. Please clarify and/or correct this sentence.

- Line 498: "Consistent in"??????

- Line 511: Please rephrase "We appreciate". Maybe... We thank.

- Line 532: Italics in species names, please.

- Line 535: Hawaii with capital H.

-Line 564: motA and motB in italics, please.

- Lines 572 and 576: You have to make a distinction between the two "Hu et al. 2022" here and in the text.

- Lines 689 and 695: You have to make a distinction between the two "Zhang et al. 2018" here and in the text.

3 The Supplementary material

- Fig. S1. Please indicate units on the Y axis.

- Fig. S2. I suggest "Disease symptoms following infection by strains FZ06 and MS2 in various dicot and monocot hosts."

- Fig. S3. Please consider changing the internal title of the figure for something like "Function by COG classification".

- Table S1.  Please include “Average Nucleotide Identity” in the title and not only its acronym. This table would be improved if species names are included and entries are limited to those hits equal to or higher than 95%.

- Table S4. Amplicon is the name of what we obtain (DNA) after PCR amplification of specific DNA segments called genes. Please name the column labeled "Amplicon" as 'gene', 'gene sequence' or 'gene segment' and eliminate the word 'detection in all the Table's cells.

Author Response

Response to Reviewer 2

1 Overall evaluation:

I believe that this research has sufficient merit to be published. The biological problem and the organism under analysis are important and the results provide new information useful for understanding the species richness of the genus Dickeya, together with important differences in terms of virulence between its members. The methodology was adequate and the authors presented well-supported conclusions. I have two main concerns that I would like the author of the article to address in the manuscript:

1)  Why, if the recognition of a new subspecies of D. dadantii is proposed, all comparisons are made with D. zeae and not with the first. It was very interesting to see the results related to the comparison of both species, and now we know well how the new subspecies differ from D. zeae but not from D. dadantii.

>> Thanks for this comment. First of all, the strain FZ06 we obtained was isolated from banana from the Philippines, so the experiment was designed to compare banana strains from different places at the beginning. Secondly, after identification of FZ06, we confirmed that it can be classified as D. dadantii. D. dadantii strains isolated from banana are ideal for comparison with FZ06, but unfortunately, we did not have strains of D. dadantii isolated from banana in our collection. Finally, the purpose of this study is to illustrate the genetic traits responsible for the virulence differentials between two different geographic banana strains. Our study revealed the importance of the major virulence factors in the pathogenesis of Dickeya on banana.

2) The proposal of a new subspecies deserves that the authors spend some time justifying it. Or is it enough to mention only in the Results section that a specific ANI value is sufficient to taxonomically reach such a conclusion? If that's the case, the article loses most of its merits. Present ALL the criteria and facts that you think justify the taxonomic creation of this subspecies.

>> Thanks for the comment. For the establishment of a novel species or subspecies, biochemical methods such as BIOLOG and fatty acid analysis, as well as molecular genetic analysis have always been considered as the criteria in traditional cognition. However, with the continuous development of genome sequencing technology, whole genome information of strains has become easier to be obtained, which can directly reflect the genetic evolution relationship between strains, and in my opinion, the essential homology between genetic materials of strains should be regarded as the gold standard of species classification. After all, the deviations of results caused by physiological, biochemical and chemotaxonomic methods due to operation and environmental factors cannot be ignored.

I think that with the development of technology, this view is more and more widely accepted. International Journal of Systematic and Evolutionary Microbiology is the main forum for the publication of novel microbial taxa, and in 2021 they presented a proposal for the description of new species as follows "Chemotaxonomic methods played an important role in the development of the polyphasic approach to classification of Archaea and Bacteria. However, we here argue that routine application of these methods is unnecessary in an era when genomic data are available and sufficient for species delineation. Thus, authors who choose not to utilize such methods should not be forced to do so during the peer review and editorial handling of manuscripts describing novel species." Some of the original articles and links are attached below: https://www.microbiologyresearch.org/content/journal/ijsem/10.1099/ijsem.0.005127; https://www.microbiologyresearch.org/content/journal/ijsem?page=about-journal.

We hope that the work we have done on the phenotypic and genomic analysis on FZ06 will support our proposal for a new subspecies of D. dadantii. If you do not agree with this, we can also accept removing the proposal of a new subspecies, because originally this article was aimed to compare the genetic and pathogenic differences between two banana strains from different countries.

2 The manuscript:

The article is easy to read, but sometimes one finds shocking sentences that all derive from inaccurate use of language. Details are provided below. However, after making all the suggested corrections, ask a person whose level of English is native to proofread the manuscript.

>> We greatly appreciate the reviewer for many helpful and constructive comments. We have followed the reviewer’s suggestions to revise the manuscript carefully in terms of language, text organization, and presentation.

Specific comments:

- Line 11: ..."FZ06 belongs to D. dadantii"... belongs????

>> We have replaced “belongs” by “can be classified”.

- Line 20: delete "Findings in" and rephrase "...clarified the genetic differentiation of Dickeya interspecific strains associated with pathogenicity...." because it makes no logical sense in the way it is written.

>> Thanks for the suggestion. We have deleted this sentence.

- Line 42: Change "and brought" to "that brought".

>> Changed as suggested.

-Line 46: Current name Streptocarpus ionanthus.

>> Thanks for the suggestion. We have updated the host name as suggested.

-Line 56: The accepted name is Canna indica.

>> Revised as suggested.

-Line 75: "belongs to D. dadantii".... belongs????

>> We have replaced “belongs” by “can be classified”.

-Line 78: change "banana" to "bananas".

>> Revised as suggested.

- Lines 78/79: Change "of the two strains" to "of the two strains MS2 and FZ06".

>> Revised as suggested.

- Line 86: Reference? LB is a well-known media and recipe but it was devised and created by somebody...

>> We are sorry for the mistake. The correct recipe and the reference have been provided.

- Line 104: change the title of the protocol to something like "Phylogenetic and genomic analyses" since the current title is a result.

>> Changed as suggested.

- lines 118/123: References for all these methods and programs are missing.  

>> We have added references to these methods in the manuscript.

- Line 132: Winter squash is Cucurbita maximaBenicasa hispida is the scientific name of what is known as the wax gourd. So, here... Which one was it?

>> Thanks for the comment. The inoculum material here is wax gourd, and we changed the “winter squash” to “wax gourd”.

-Line 132: The correct designation of Chinese cabbage is Brassica rapa subsp. pekinensis.

>> Changed to Brassica rapa subsp. pekinensis.

- Line 135: the scientific name of tomato is Solanum lycopersicum.

>> Changed to Solanum lycopersicum.

- Line 138: this was NOT a negative control but a 'Blank'.

>> Changed as suggested.

- Line 141: Please rephrase "pictured till diseased symptoms were observed". What's that supposed to mean???

>> We have changed "in the figure until disease symptoms are observed" to "until symptoms appear".

- LInes 145/151: Please provide references or state that all those methods were developed in your lab.

>> We have added references to the manuscript.

- Lines 153/159:  I guess there are references (at least one) for all these media and methods... aren't they?

>> We have added the reference in the text.

- Line 166: Explain "supernatants were discarded, dried at 55°C and weighed" better (i.e., you dried what was discarded?) and provide a reference for the method.

>> Thank you for your comment. We have reinterpreted this statement as "supernatants were discarded, and the precipitate was dried and weighed at 55°C" and have included a reference in the manuscript.

- Line 170: Change "Trpetone" to "Tryptone".

>> Corrected.

- Lines 169/177: Please provide references.

>> We have added the reference in the text.

- Lines 199/200: Change "protein-coding genes, 75 tRNAs, 9 ncRNAs, and 22 rRNAs..." to "protein-coding, 75 tRNAs, 9 ncRNAs, and 22 rRNAs genes..."

>> Changed as suggested.

- Line 206: Please check the correct use of plurals (e.g., pseudogenes, not pseudogene, and so on) in Table 1

>> We checked carefully and corrected the use of plurals in Table 1.

- Line 217: I prefer you use "relatedness analysis" instead of phylogeny (which is not).

>> Changed as suggested.

- Lines 224/227: If this is really relevant please move it to the Discussion section and explain why it is important.

>> Thank you for your comment. We have moved this section to the discussion and expanded the content as suggested. As follows: "During analyzing the relatedness between strain FZ06 and the near strains, we found that FZ06, S3-1, and NCPPB 2976 were isolated from diseased samples of monocotyledonous hosts (A622-S1-A17 from river water), while almost all other D. dadantii strains were isolated from dicotyledonous hosts (Table S2). Notably, the three strains of D. dadantii derived from monocotyledonous hosts belong to the same sister branch in the taxonomic status of the evolutionary tree (Fig. 4), and they have a closer evolutionary relationship. This finding helps to confirm the identity of the new subspecies of FZ06 (isolated from monocotyledonous hosts with NCPPB 2976 as a subspecies of D. dadantii), and their distribution on monocotyledonous host plants may be related to the bacterial evolution and host specificity during plant-pathogen interactions".

- Line 229: I would suggest changing the title of this Figure.

>> Revised as “Phylogenetic tree based on 1,264 bacterial conserved orthologous single copy genes of strain FZ06, 16 D. dadantii and 11 type strains of other Dickeya spp.”

- Line 232: Please consider sending Table 2 to Supplementary material.

>> We have followed the reviewer’s suggestion and moved Table 2 to supplementary Table S2.

- Lines 243-246: Please check the correct use of taxonomic names taking into consideration the observations made before.

>> Thanks to your suggestions, we have carefully revised the manuscript and corrected all incorrect species names.

- Line 247: Please change the title of this Table to something more accurate and informative. For example, What are those values? Lesions sizes?

>> Thanks for the comment. We have change the title more accurate as “Lesion sizes or weights on 14 dicotyledonous and monocotyledonous plants caused by strains FZ06 and MS2”.

- Lines 251-252: Please rephrase, it is confusing. Make emphasis on the virulence differences among strains; in the way the sentence is written, it seems that you point to differences among the virulence factors themselves.

>> Rephrased as “We measured multiple virulence factors produced by strains FZ06 and MS2 to determine the differences in virulence between the strains”.

- Line 265: Please change "till pictures captured" to "until photographed".

>> Revised as suggested.

- Lines 267 and 273: change "performed" to "perform".

>> Revised as suggested.

- Line 394: Please rephrase "crop, food, vegetable, and flower industries." What is a crop industry?, for example.

>> Thank you for your comment. We have revised the sentence as “grain, vegetable, and flower production”.

- Line 430: What is "Although the virulence of PelA and PelN infestation was relatively low" suppose to mean? To me, this sentence does not make sense at any level.

>> Thank you for your comment, we removed the phrase.

- Line 458: change "part due" to "in part due".

>> Revised as suggested.

- Line 465: Change "film (pellicle)" to "biofilm".

>> Revised as suggested.

- Line 478: This is confusing and misleading: you are talking about Xanthomonas spp and then jump to mention X. fastidiosa. Are you referring to Xyllela fastidiosa? Because that's a different bacterial genus. Furthermore, the reference provided is about Xanthomonas oryzae pv. oryzae. Please clarify and/or correct this sentence.

>> We wanted to express that there have been in-depth studies on pilMNOPQ operon in a variety of Gram-negative bacteria (e. g. Pseudomonas aeruginosa and Xanthomonas spp.) and cited the results of the research on Xyllela fastidiosa. We apologize for the confusion, and we have rephrased the sentence and added the appropriate references as “The pilMNOPQ operon encodes a key T4P structural component that is essential for pilus biogenesis. Studies on the pilMNOPQ operon gene are mostly found in Pseudomonas aeruginosa and Xanthomonas spp. (Lim et al., 2008; Koo et al., 2013). In addition, it has been shown that pilQ mutants of Xyllela fastidiosa could not twitch (Meng et al., 2005)”.

- Line 498: "Consistent in"??????

>> We have revised the sentence as “T4SS-deficient strains are common in Pectobacterium and Dickeya spp. and the copy numbers of T4SS vary between strains”.

- Line 511: Please rephrase "We appreciate". Maybe... We thank.

>> Revised as suggested.

- Line 532: Italics in species names, please.

 >> Revised as suggested.

- Line 535: Hawaii with capital H.

>> Corrected.

-Line 564: motA and motB in italics, please.

>> Revised as suggested.

- Lines 572 and 576: You have to make a distinction between the two "Hu et al. 2022" here and in the text.

>> Thank you for the suggestion, we have added the letters a and b after the year to distinguish them.

- Lines 689 and 695: You have to make a distinction between the two "Zhang et al. 2018" here and in the text.

>> Thank you for the suggestion, we have added the letters a and b after the year to distinguish them.

3 The Supplementary material

- Fig. S1. Please indicate units on the Y axis.

>> Revised as suggested.

- Fig. S2. I suggest "Disease symptoms following infection by strains FZ06 and MS2 in various dicot and monocot hosts."

>> Revised as suggested.

- Fig. S3. Please consider changing the internal title of the figure for something like "Function by COG classification".

>> Revised as suggested.

- Table S1.  Please include “Average Nucleotide Identity” in the title and not only its acronym. This table would be improved if species names are included and entries are limited to those hits equal to or higher than 95%.

>> Thank you for your comments. We have included “Average Nucleotide Identity” in the title and added the species name of the strains in the table list. The full list not only provides the strains with hits equal to or higher than 95%, but also other strains classified in other Dickeya species. We believe that these data are convincing for the classification of strain FZ06 and insist to keep in Table S1.

- Table S4. Amplicon is the name of what we obtain (DNA) after PCR amplification of specific DNA segments called genes. Please name the column labeled "Amplicon" as 'gene', 'gene sequence' or 'gene segment' and eliminate the word 'detection in all the Table's cells.

>> Thank you for your suggestion. We have changed it as suggested.

Reviewer 3 Report

The Results and Discussion section can be modified and presented in a more systematic style. The discussion section should not be repetitive to the result section but have a clear and concise discussion on the results and be supplemented with supporting references.

Overall, the work bears merit.

Author Response

The Results and Discussion section can be modified and presented in a more systematic style. The discussion section should not be repetitive to the result section but have a clear and concise discussion on the results and be supplemented with supporting references.

Overall, the work bears merit.

>> Thanks very much for your comments. We have carefully revised the sections of Results and Discussion, and enriched the discussion with the help of more supporting references.

Round 2

Reviewer 1 Report

When describing genome sequences in publications, quote the accession number, not the bio project number.
I therefore kindly ask the authors to quote the correct accession number for strain FZ6, and the MS2 strain should also cite the accession number, i.e. CP025799.

Author Response

  1. When describing genome sequences in publications, quote the accession number, not the bio project number. I therefore kindly ask the authors to quote the correct accession number for strain FZ6, and the MS2 strain should also cite the accession number, i.e. CP025799.

>> Thank you for your comments. At the time of your previous suggestion, the GenBank accession number for FZ06 had not yet been published, so we used the biological project number. Now the genome information has been released, and we have replaced the project number in the manuscript by the GenBank accession number as follows: FZ06: CP094943.1; MS2: CP025799.1.